# Promise and Challenges of T Cell Immunotherapy for Osteosarcoma

**DOI:** 10.3390/ijms241512520

**Published:** 2023-08-07

**Authors:** Jeong A Park, Nai-Kong V. Cheung

**Affiliations:** 1Department of Pediatrics, Inha University College of Medicine, Incheon 22212, Republic of Korea; 2Pediatrics, Memorial Sloan Kettering Cancer Center, New York, NY 10065, USA; cheungn@mskcc.org

**Keywords:** chimeric antigen receptor, immune checkpoint inhibitors, myeloid-derived suppressor cells, osteosarcoma, T cell engaging bispecific antibody, T cell immunotherapy, tumor microenvironment, tumor-associated macrophage, vascular endothelial growth factor

## Abstract

The cure rate for metastatic or relapsed osteosarcoma has not substantially improved over the past decades despite the exploitation of multimodal treatment approaches, allowing long-term survival in less than 30% of cases. Patients with osteosarcoma often develop resistance to chemotherapeutic agents, where personalized targeted therapies should offer new hope. T cell immunotherapy as a complementary or alternative treatment modality is advancing rapidly in general, but its potential against osteosarcoma remains largely unexplored. Strategies incorporating immune checkpoint inhibitors (ICIs), chimeric antigen receptor (CAR) modified T cells, and T cell engaging bispecific antibodies (BsAbs) are being explored to tackle relapsed or refractory osteosarcoma. However, osteosarcoma is an inherently heterogeneous tumor, both at the intra- and inter-tumor level, with no identical driver mutations. It has a pro-tumoral microenvironment, where bone cells, stromal cells, neovasculature, suppressive immune cells, and a mineralized extracellular matrix (ECM) combine to derail T cell infiltration and its anti-tumor function. To realize the potential of T cell immunotherapy in osteosarcoma, an integrated approach targeting this complex ecosystem needs smart planning and execution. Herein, we review the current status of T cell immunotherapies for osteosarcoma, summarize the challenges encountered, and explore combination strategies to overcome these hurdles, with the ultimate goal of curing osteosarcoma with less acute and long-term side effects.

## 1. Introduction

Bone sarcomas represent about 6% of all pediatric cancers, of which osteosarcoma makes up the majority (56%), making it the most common primary bone malignancy for children and young adults. The patients diagnosed with metastatic or relapsed osteosarcoma still have dismal outcomes despite multimodal treatment approaches such as conventional multi-agent chemotherapy, surgery, or high-dose chemotherapy with stem cell transplantation, achieving long-term survival in less than 30% of cases [1]. Moreover, given the young age of onset for osteosarcoma, the side effects of these treatments can be devastating and long-lasting. Even patients in remission can suffer from long-term complications including secondary malignancies, disfigurement (from surgery), and psychosocial trauma [2,3]. As such, there is a desperate need for more effective and less toxic therapy for both localized and metastatic high-risk osteosarcoma. Immunotherapy may offer viable alternatives. Since the reports of Dr. Coley on bacterial toxins inducing tumor regression [4], many immunotherapy attempts have been made in soft tissue and bone sarcomas, but so far without consistent or durable response [5,6]. Although interferons (IFN) are well known to have anti-angiogenic, anti-tumor, and immune-stimulating properties [5], the EURAMOS-1 clinical trial incorporating IFN-α2b as a maintenance therapy failed to show clinical benefit [7]. Monoclonal IgG antibodies targeting specific tumor surface antigens have also been tested, including trastuzumab to target HER2 (human epidermal growth factor receptor 2) [8], cetuximab to target epidermal growth factor receptor (EGFR) [9], glembatumumab vedotin (GV) to target glycoprotein nonmetastatic B (gpNMB) [10], denosumab to target the cytokine RANKL (receptor activator of NFκB ligand) (NCT02470091), and dinutuximab to target disialogangliosides (GD2) (NCT 02484443), but anti-tumor effects have been transient or inconsistent [11,12,13]. Recently, trastuzumab deruxtecan (an antibody-drug conjugate consisting of trastuzumab and topoisomerase I inhibitor deruxtecan) was FDA approved in patients with HER2-low metastatic breast cancer [14]. A phase 2 study of trastuzumab deruxtecan is ongoing for the treatment of HER2(+) osteosarcoma (NCT04616560), but the preliminary results are disappointing: seven out of eight patients showed progressive disease, while one showed a stable disease [15].

T cell immunotherapy has proven activity for many high-risk malignancies, but their efficacy against osteosarcoma remains largely unexplored. Although preclinical studies using immune checkpoint inhibitors (ICIs), antigen-specific chimeric antigen receptor (CAR), or bispecific antibody (BsAb) have demonstrated the impressive anti-tumor capacity of T cells, immunosuppressive tumor microenvironment (TME) remains a major barrier. Bone tumors, including osteosarcomas, grow in a bone microenvironment, unique among primary tumors while common for metastases with preference for the bony niche. This TME is composed of a variety of cells including bone cells (osteoblasts, osteoclasts, osteocytes), stromal cells (mesenchymal stromal cells, fibroblasts), vascular cells (endothelial cells and pericytes), immune cells (dendritic cells (DCs), T cells, tumor-associated macrophages (TAMs), myeloid-derived suppressive cells (MDSCs), and NK cells), and a mineralized extracellular matrix (ECM). Cross-talks between osteosarcoma and the TME are channeled through diverse environmental signals such as cytokines, chemokines, and soluble growth factors [13] that promote tumor growth and metastasis while simultaneously thwarting immune surveillance. This osteosarcoma-specific TME impedes T cell infiltration into tumors, accelerates immune effector cell exhaustion and anergy, and derails anti-tumor immunity, creating both a major roadblock and a potential tumor vulnerability.

Herein, we review the promise and the limitations of T cell immunotherapies for osteosarcoma, focusing on ICIs, BsAbs, and CAR T cells, and osteosarcoma-specific TME. We explore strategies to overcome the immune-hostile TME and combination approaches to create synergy with T cell immunotherapy for osteosarcoma.

## 2. Immune Checkpoint Inhibitors for Osteosarcoma

Upregulation of programmed cell death-1 receptor (PD-1) on CD8(+) T cells promotes T cell exhaustion and dysfunction in chronic inflammation [16,17,18,19]. PD-1 and tumor PD-L1 interaction promotes T cell tolerance through suppressing release of immunostimulatory cytokines while directly inhibiting T cell cytotoxicity [20]. ICIs reverse this process by reinvigorating cytotoxic T lymphocytes (CTLs), reviving immune response directed at neoantigens distinct from those on host tissues [21,22]. Despite the low tumor mutational burden (TMB) in pediatric cancers in general, neo-epitopes arising from genetic instability in osteosarcoma could offer potential targets for T cell-mediated cytotoxicity, potentially exploitable by ICI therapies [23,24,25].

After T cell receptor (TCR) activation, cytotoxic T-lymphocyte-associated protein 4 (CTLA4) (CD152), type I transmembrane glycoprotein, is upregulated and constitutively expressed on CTLs and regulatory T cells (Tregs), and after binding to CD80 and CD86 with higher affinity and avidity than CD28 results in T cell suppression and DC dysfunction [26,27,28,29]. Blockade of the CTLA4 receptor increased the number of CD8(+) T cells while reducing Tregs, and combination with tumor lysate-loaded DC inhibited metastasis and prolonged survival of mice with fibrosarcoma [30].

PD-1 is also expressed on T cells following TCR engagement and activation. PD-1 and PD-L1 ligation exerts inhibitory signals for T cell activation (Figure 1) [29]. Overexpression of PD-1 and PD-L1 and their interactions are well-characterized immune escape mechanisms of osteosarcoma [29,31,32]. Besides the direct inhibition of effector T cells, PD-1/PD-L1 interactions reduce the capacity of CD4(+) T cells to secrete IL-21 necessary for CTL response [33] and affect cytotoxicity of NK cells by reducing granzyme B secretion [34]. PD-1 was increased in circulating T cells in osteosarcoma patients, and PD-L1 expression in osteosarcoma was related to early metastasis and poorer outcome [32,35,36]. While PD-L1 density in osteosarcoma cell lines varies widely from low to high, the drug-resistant variants trend towards higher values compared to their parent counterparts [37]. A study using CRISPR/Cas9 system to target the PD-L1 gene in osteosarcoma cells revealed that PD-L1 regulates osteosarcoma growth and drug resistance [38]. The expression levels of PD-L1 correlated with TILs [37], and the blockade of PD-1/PD-L1 interactions improved the activity of osteosarcoma-reactive CTLs, resulting in an improved outcome in preclinical models [39,40]. PD-1 inhibitor could effectively control osteosarcoma pulmonary metastasis by increasing CD4(+) and CD8(+) TILs and enhancing the cytolytic activity of CD8(+) T cells in the lung [41]. Both human and murine metastatic osteosarcomas express the PD-L1, which could functionally impair tumor-infiltrating CTLs by engaging their surface PD-1. This model was supported by studies where the PD-L1 blockade improved the function of osteosarcoma TILs in vivo, decreasing tumor burden and increasing survival of mice carrying metastatic osteosarcoma [39]. The combination of triple antibodies, anti-PD-1, anti-PD-L1, and anti-OX-40 agonistic antibody, led to a prolonged survival of mice in preclinical studies, suggesting the therapeutic potential of the PD-1/PD-L1 pathway for high-risk osteosarcoma [40].

However, clinical studies of ICIs have failed to produce satisfactory results in osteosarcoma. Phase I study of ipilimumab in pediatric patients with advanced or relapsed solid tumors including eight osteosarcomas failed to show clinical benefit as a single agent, despite observing an increase in activated HLA-DR(+) Ki67(+) T cells without concomitant upregulating Tregs among the patients [42]. A recent phase II clinical trial of anti-PD1 pembrolizumab for advanced sarcomas reported that 7 out of 40 patients with soft tissue sarcoma (18%) and only 2 out of 40 patients (5%) with bone sarcomas had objective responses [43]. The study included twenty-two patients with osteosarcoma; one patient (5%) had a partial response, six patients (27%) had a stable disease, and fifteen patients (68%) showed disease progression. Another study of pembrolizumab in advanced osteosarcoma also failed to show clinical benefit despite high PD-L1 expression in tumors (11 of 12 patients): median progression-free survival (PFS) was 1.7 months and median overall survival (OS) was 6.6 months [44]. A clinical trial of the PD-L1 inhibitor (avelumab) for recurrent or progressive osteosarcoma was no more successful, where 17 out of the 18 treated patients showed disease progression while on study (NCT03006848). The low clinical activity of single PD-1 or PD-L1 blockade in most sarcoma subtypes suggests that PD-1 or PD-L1 inhibitor alone cannot adequately revive exhausted or tolerized effector T cells in these patients. These results contrast with undifferentiated pleomorphic sarcomas showing good clinical response accompanied by high numbers of TILs [43], emphasizing the need to develop strategies to enhance T cell infiltration. Although the combination of two ICIs acting through different mechanisms, such as anti-CTLA4 and anti-PD-1, has shown synergy in preclinical models of osteosarcoma as well as in those of melanoma [40,45], such combinations have had mixed response so far in bone sarcomas. A combination of nivolumab and ipilimumab failed to show efficacy in patients with osteosarcoma [46], and a combination of durvalumab (anti-PD-1) and tremelimumab (anti-CTLA4) resulted in two partial responses out of five osteosarcoma patients treated [47]. Several cases reported that the combination of anti-CTLA4 and anti-PD-1 antibodies induced remission and tumor stabilization in patients with metastatic osteosarcoma [48,49], while the addition of camrelizumab (anti-PD-1 inhibitor) to the inhibition of vascular endothelial growth factor receptor 2 (VEGFR2)] using apatinib (tyrosine kinase inhibitor (TKI)) was shown to prolong PFS of patients with advanced osteosarcoma compared with apatinib alone [50]. The overall findings suggest that a combination strategy rather than a stand-alone therapy may be the path to the future. The data on clinical trials of ICIs for osteosarcoma are summarized in Table 1.

## 3. Adoptive T Cell Immunotherapy for Osteosarcoma

While ICIs are nonspecific, adoptive T cell immunotherapy (ATC) using CAR or BsAb is tumor-antigen-specific, directly driving T cells to the tumors and inducing potent cytotoxicity. CAR and BsAb are engineered to recognize tumor-associated antigens (TAAs) and exert T cell-mediated cytotoxicity in a major histocompatibility complex (MHC)-independent manner. Although CAR T cells achieved extraordinary clinical success in hematologic malignancies receiving FDA approvals, they exhibit generally inconsistent and non-durable effects on solid cancers due to tumoral heterogeneity, physical barrier, aberrant vasculature, and the immunosuppressive TME [55,56,57,58]. Anti-tumor efficacy of ATC primarily derives from recognition by VH (variable region of heavy chain) and VL (variable region of light chain) of target antigen-specific antibodies. Ideal TAAs carry epitopes exclusively expressed on tumor cell surface to allow engineered antibodies or receptors to drive T cells selectively into tumors while minimizing off-tumor toxicities. To avoid tumor escape, TAA should be ideally expressed homogenously within and between tumors among patients. The targets reported so far for ATC in osteosarcoma include HER2, GD2, B7-H3 (CD276), interleukin-11 receptor α-chain (IL-11Rα), insulin-like growth factor 1 receptor (IGF1R), receptor tyrosine kinase-like orphan receptor 1 (ROR1), erythropoietin-producing hepatocellular class A2 (EphA2), natural killer group 2D ligand (NKG2DL), activated leukocyte cell adhesion molecule (ALCAM, CD166), folate receptor-α (FRα), chondroitin sulfate proteoglycan 4 (CSPG4), and CD151 [59,60,61]. Among them, HER2, GD2, and B7H3 have been studied the most for osteosarcoma [62,63,64,65,66,67].

Although osteosarcoma cell lines and tissue sections were HER2-positive by immunohistochemistry or flow cytometry [11,62], HER2 gene amplification was rarely observed in osteosarcoma, and the expression levels were much lower than those of HER2(+) breast cancers [68], accounting for the clinical insensitivity of OS to trastuzumab [8]. Despite its HER2 antigen density being too low for conventional IgG-mediated cytotoxicity, osteosarcoma was effectively killed by HER2-CAR T cells, which, when injected intratumorally, induced the regression of established osteosarcoma xenografts, prolonging survival of the mice [11]. HER2-CAR T cells also decreased the sarcosphere forming capacity and bone tumor generating ability, suggesting the potential to target osteosarcoma stem cells [65]. A phase I/II study of HER2-CAR T cells without lymphodepletion resulted in a stable disease in 3 and progressed disease in 12 among 16 patients with recurrent/refractory HER2(+) osteosarcoma (NCT00902044). Although these results were modest, HER2-CAR T cells could traffic to tumor sites and persist for more than 6 weeks in a dose-dependent manner [64]. In the same trial, five osteosarcomas, three rhabdomyosarcomas, one Ewing sarcoma, and one synovial sarcoma were treated with HER2-CAR T cells following lymphodepletion. Among them, three had a stable disease, five had a progressive disease, while one rhabdomyosarcoma and one osteosarcoma patient had complete remission for 12 months and 32 months, respectively [69].

GD2, another promising target for CAR and BsAb, is overexpressed in many cancers including osteosarcoma while being limited in normal tissues [62,70,71]. GD2 has a role in signal transduction and cell adhesion, and overexpression of GD2 increases the phosphorylation of paxillin and focal adhesion kinase (FAK), promoting migration and invasion of osteosarcoma cells [72]. The third-generation GD2-CAR T cells using anti-GD2 clone 14G2a successfully recognized GD2(+) sarcoma cell lines and showed cytotoxicity in vitro, but the accumulation of myeloid-derived suppressor cells (MDSCs) attenuated the anti-tumor effect of GD2-CAR T cells in vivo [73]. This inhibition phenomenon was observed in a clinical trial of GD2-CAR T cells in neuroblastoma: GD2-CAR T cells with or without lymphodepletion resulted in modest antitumor responses, with a striking expansion of CD45/CD33/CD11b/CD163 (+) myeloid cells in all patients [74]. The fourth-generation GD2-CAR T cells using the hu3F8 clone can also effectively target osteosarcoma cells and induce PD-L1 on tumor cells and PD-1 on GD2-CAR T cells, limiting T cell activity. Combination with low-dose doxorubicin decreased PD-L1, enhancing the potency of GD2-CAR T cells on osteosarcoma in vitro [66]. Recently, Del Bufalo et al. reported exceptional results of GD2-CAR T cells in patients with relapsed or refractory neuroblastoma. Twenty-seven patients were treated with third-generation GD2-CAR T cells, and the overall response was 63%; nine patients had a complete response, eight had a partial response; toxicities were tolerable, and the inducible caspase 9 suicide gene was needed only in one patient (GD2-CART01) [75]. Another third-generation GD2-CAR T cells combined with a safety switch (GD2-CAR.OX40.28.z.ICD9) are being tested for solid tumors including osteosarcoma in a phase I clinical trial (NCT02107963).

B7-H3 (CD276) CAR T cells are also being tested for the treatment of osteosarcoma. B7-H3 is a checkpoint molecule expressed at high levels on pediatric solid tumors including osteosarcoma [76,77], and it contributes to tumor immune evasion and metastasis, correlating with poor prognosis [78]. B7-H3-targeting CAR T cells have shown anti-tumor activity in osteosarcoma xenograft models [67,79]. A phase I clinical trial is currently recruiting patients with solid tumors that express B7-H3 (NCT04483778).

On the other hand, T-BsAb represents another promising alternative which effectively drives T cells to the tumor sites with less toxicity. T-BsAbs are also mostly known for their use in hematological malignancies similarly to CAR T cells, and blinatumomab (a CD3 × CD19 BsAb built on scFv framework) was FDA approved in 2014 and has been successful against relapsed or refractory acute lymphoblastic leukemia (ALL) [80,81]. For solid tumors, catumaxomab (CD3xEpCAM BsAb) targeting epithelial cell adhesion molecule (EpCAM)-positive cancers showed benefit in reducing malignant ascites secondary to epithelial cancers, with an acceptable safety profile [82,83,84]. To improve efficacy and to reduce clinical toxicities, BsAb-armed T cells, using chemically conjugated BsAb, anti-GD2 × anti-CD3 [hu3F8 × mouse OKT3 (NCT02173093)], anti-HER2 × anti-CD3 [trastuzumab × mouse OKT3 (NCT00027807)], or anti-EGFR × anti-CD3 [cetuximab × mouse OKT3 (NCT04137536)], were developed and tested in clinical studies, proven effective and safe in breast, neuroblastoma, prostate, and pancreatic cancers [85,86,87,88]. GD2-BsAb-armed T cells were tested for their efficacy on GD2-positive tumors and induced a significant PET response in one out of three osteosarcoma patients (NCT02173093) [89]. To harness the potential of BsAb against solid tumors, the BsAb structural format was found to be critical [90,91]. Despite similar in vitro anti-tumor properties of GD2-BsAb formats, including monomeric BiTE, dimeric BiTE, IgG heterodimer, IgG-[H]-scFv, or chemical conjugate, the IgG-[L]-scFv format, where the anti-CD3 (huOKT3) scFv was attached to the light chain of a tumor binding IgG, proved the most effective in vivo, driving more T cells into tumors and producing more durable anti-tumor responses [91]. For osteosarcoma cell lines that are HER2-positive and/or GD2-positive, IgG-[L]-scFv GD2-BsAb (hu3F8 × huOKT3) or HER2-BsAb (trastuzumab × huOKT3) administered intravenously successfully drove T cells into tumors to exert potent cytotoxicity in vivo [62]. T cells armed ex vivo (EAT) with the IgG-[L]-scFv-formatted GD2-BsAb (GD2-EATs) or HER2-BsAb (HER2-EATs) also successfully ablated both osteosarcoma cell-line-derived xenografts (CDXs) and patient-derived xenografts (PDXs) with significantly lower cytokine release while increasing the overall survival [62,91]. Although a phase I/II study of this IgG-[L]-scFv-formatted GD2-BsAb (Nivatrotamab) in patients with relapsed/refractory neuroblastoma, osteosarcoma, and other GD2(+) solid tumors was temporarily suspended because of company business priorities (NCT03860207), clinical results are anticipated. The results of clinical trials of BsAb or CAR T cell therapy for osteosarcoma conducted to date are summarized in Table 2.

## 4. Tumor Microenvironment in Osteosarcoma

Despite the excitement regarding the clinical utility of T cell immunotherapy, the overall response rate of ICIs is around the 20% range across solid tumors [93], where the promise of ATC remains elusive as well. The challenges of successful T cell immunotherapy in osteosarcoma include poor immunogenicity, paucity of neoantigens and TILs, obvious tumor heterogeneity, and the osteosarcoma-specific, immunosuppressive TME [94].

As a seed, osteosarcoma flourishes in soil called the TME. The TME of osteosarcoma consists of a special, complex, and highly structured osseous environment, populated by osteocytes, stromal cells, vascular cells, immune cells, and mineralized architecture, ECM [95] (Figure 1). Bone matrix remodeling is a unique feature of osteosarcoma. The activation of the RANK-RANKL signaling pathway leads to osteoclast activation, resulting in excessive bone resorption and the release of bone matrix growth factors such as transforming growth factor-β1 (TGFβ1), insulin-like growth factor-1 (IGF1), fibroblast growth factor (FGF) or bone morphogenetic protein (BMP), which in turn promote tumor cell proliferation and further bone destruction [96]. These growth factors not only inhibit osteogenic differentiation but also prohibit T cell proliferation and differentiation, sabotaging host immune surveillance [97]. Bone marrow mesenchymal stem cells (MSCs) secrete cytokines, chemokine ligand 5 (CCL5), IL-6, and VEGF that promote growth, metastasis, and angiogenesis of osteosarcoma, while the MSC-derived osteoblasts deposit and mineralize the ECM [98]. This osteoid tumor matrix not only fuels tumor growth and metastasis but also limits the trafficking and infiltration of T cells, playing a role as an ideal milieu for osteosarcoma progression while putting up a potentially insurmountable barrier to T cell immunotherapy [99,100].

**Figure 1 ijms-24-12520-f001:**
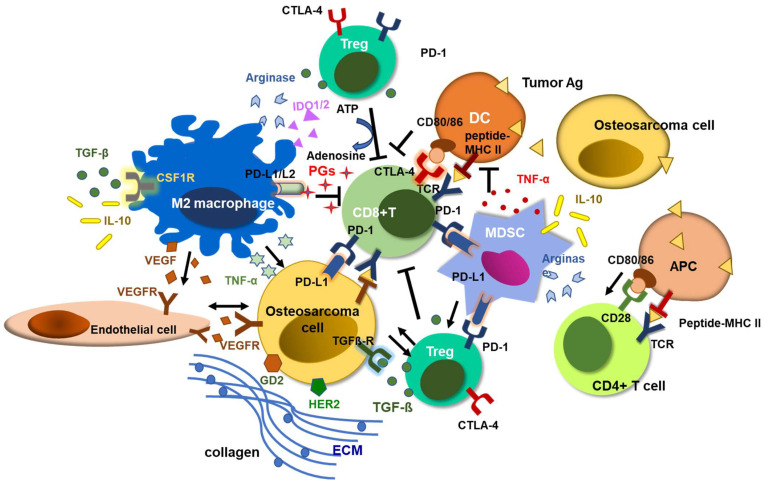
Tumor microenvironment of osteosarcoma. Osteosarcoma tumor tissues have large populations of tumors infiltrating myeloid cells (TIMs), including MDSCs and TAMs. Immunosuppressive TIMs and dense ECM around osteosarcoma cells impede T cell infiltration and cytotoxic activity. Tumor infiltrating T cells express PD-1, and the interactions between PD-1 and PD-L1 expressed on tumor cells and TIMs exhaust cytotoxic CD8(+) T cells, inducing tumor-immune tolerance. Regulatory T cells (Tregs) release TGF-β and convert ATP to adenosine via CD39 and CD73, inhibiting T cell cytotoxicity and promoting tumor progression [101]. M2 polarized TAMs and MDSCs also release immune-suppressive cytokines and chemokines including TGFβ, TNF-α, IL-10, prostaglandins, arginase, and VEGF, inducing osteosarcoma progression and immune evasion [102].

Targeting the building blocks and the regulatory elements of the ECM has been explored for osteosarcoma. These targets include collagens, fibronectin, laminins, and proteoglycans [103]. Losartan, the angiotensin II receptor blocker, inactivates cancer-associated fibroblasts (CAFs), decreases stromal collagen and hyaluronan production, and reduces TGFβ1, connective tissue growth factor (CCN2, cellular communication network factor 2), and endothelin-1 (ET-1), thereby lowering mechanical compression in tumors and increasing vascular perfusion [104]. Losartan treatment leads to a dose-dependent reduction in stromal collagen in desmoplastic models of human breast, pancreatic and skin cancers, and enhanced the efficacy of chemotherapy in multiple cancer models [105,106]. It also blocks monocyte migration into osteosarcomas and shows significant benefit against canine metastatic osteosarcoma in combination with TKI toceranib [107].

Besides the mechanical stroma, activated VEGF pathway also plays a pivotal role in osteosarcoma progression. Cancer cell metabolism is characterized by an enhanced uptake and utilization of glucose [108], and the persistent activation of aerobic glycolysis in cancer cells is linked to the activation of oncogenes or loss of tumor suppressors [109,110]. Heightened metabolism of cancer cells creates a hypoxic and acidic TME, increasing the expression of hypoxic inducible factors (HIFs), VEGFs, and other pro-angiogenic factors, which promote abnormal angiogenesis contributing to chaotic tumor microvasculature [95]. In osteosarcoma, hypoxia and lactic acidosis promote highly vascularized TME, accelerate hypoxic nutrient consumption and waste accumulation, which combine to suppress CTL proliferation and activity [95]. Chemokines (CCL3 and CCL5) and other proangiogenic factors also upregulate VEGF and promote neovasculature in osteosarcoma [111,112,113], and high expression of VEGF and VEGFR2 is associated with poor prognosis [114,115,116]. Dual silencing of the VEGF and Survivin genes effectively inhibited the proliferation, migration, angiogenesis and survival of the osteosarcoma cells [117], suggesting a potential of VEGF pathway blockade as another therapeutic maneuver to salvage T cell immunotherapy against osteosarcoma.

Beyond mechanical stroma and angiogenesis, the immunosuppressive TME is another challenge. The osteosarcoma TME is mainly orchestrated by MDSCs and M2 macrophages. While immune-inflamed ‘hot’ tumors have significant numbers of CD8(+) T cells in the tumor stroma and express pro-inflammatory cytokines, responding well to T cell immunotherapies, osteosarcoma belongs to ‘cold’ tumors characterized by the paucity of TILs, accompanied by immunosuppressive tumor infiltrating myeloid cells, such as TAMs and MDSCs, as well as regulatory T cells (Tregs) [118,119,120]. Tumors secrete high levels of colony-stimulating factor 1 (CSF-1), which converts M1 macrophages (classically activated and tumoricidal) to M2 macrophages (alternately activated, tumor-promoting) along with Th2 cytokines (e.g., IL-4, IL-10, TGFβ1, and PGE2) and stimulates tumor growth and metastasis [121,122]. CD14/CD68 double-positive TAMs are the main immune infiltrates in osteosarcoma, and RNA analyses revealed that type 2 TAMs are the most abundant immune infiltrates [123]. M2 TAMs release proangiogenic factors, such as FGF, matrix metallopeptidase 9 and 12 (MMP-9, MMP-12), and VEGF to increase angiogenesis and vascular extravasation while suppressing CTLs and maintaining Tregs [124,125,126]. TAM-modulating agents including mifamurtide (MTP-PE), ATRA, metformin, gefitinib, esculetin, zoledronic acid, and CAR-macrophages have been tested in osteosarcoma with promising results in preclinical studies [127].

In addition, despite the limited efficacy of radiotherapy in treating osteosarcoma, there have been reports of immunomodulatory effects of radiation on the TME as well as its promising synergy with T cell immunity [128,129,130]. Radiotherapy has the potential to ignite tumor immune recognition by generating immunogenic signals and releasing neoantigens [130]. It triggers recruitment of CD11b (+) myeloid cells and reprogramming of macrophages toward the M2-phenotype [131], simultaneously increasing CD4(+) and CD8(+) T cells in the TME [132]. In two recent preclinical studies, 90Y-NM600 activated the STING-IFN1 signaling pathway and increased proinflammatory cytokines [133,134]. Radionuclide therapy also enhanced infiltration of immunostimulatory CD4(+) and CD8(+) T cells, APCs, NK cells, and other innate immune cells into the TME and affected TAMs and Tregs, having the potential to target TME by radioimmunotherapy [133,134,135,136]. Lutetium-177 (177Lu) is a promising therapeutic radionuclide with suitable β(-) energy and physical half-life. When targeted to mineralized bone cells, it could induce apoptotic osteosarcoma cell death while being effective for cancer bone pain palliation [137,138]. 177Lu-PSMA increased T cell infiltration into tumors and induced immunogenic cancer cell death and modulated TME, improving progression-free and overall survival in metastatic castration-resistant prostate cancer [139]. Phase 1 clinical trial of pretargeted radioimmunotherapy (PRIT) using GD2-specific Self-Assembling Disassembling (GD2-SADA) BsAb platform to deliver 177Lu-DOTA payload in patients with recurrent or refractory GD2(+) solid tumors including osteosarcoma is ongoing (NCT05130255) [140].

## 5. Combination Strategies to Overcome the Limitations of T Cell Immunotherapy against Osteosarcoma

Among the latest trends in T cell immunotherapy, various combination approaches are actively explored. First, strategies that combine ATC with ICI may encourage functional persistence of BsAb- or CAR-driven T cells in osteosarcoma. CAR or BsAb drives TILs to exert highly specific anti-tumor immune responses [62,141], which can be theoretically amplified by the addition of ICIs to reinvigorate exhausted T cells [142]. In a HER2(+) breast cancer model, HER2-CAR T cells upregulated PD-1 after incubation with target cells, and the PD-1 blockade did increase CAR T cell proliferation, IFN-γ production, and granzyme B expression in vitro while enhancing in vivo cytotoxicity [143]. The third-generation GD2-CAR T cells had highly potent immediate cytotoxicity, but significant activation-induced cell death (AICD) was observed after chronic antigen stimulation, where the PD-1 blockade enhanced GD2-CAR T cell survival and cytotoxicity against melanoma cell lines [144]. Cherkassky et al. also reported that PD-1 inhibitors rescued the effector function of exhausted mesothelin-specific CAR T cells and improved the potency of CAR T cells in a model of pleural mesothelioma [145]. However, the clinical study of GD2-CAR T cells combined with the PD-1 inhibitor failed to achieve the intended synergy; the PD-1 inhibitor did not further enhance GD2-CAR T cell expansion or persistence, laying the blame on tumor-infiltrating macrophages [74]. A phase I clinical study of HER2-CAR T cells in combination with PD-1 antibody to test safety and efficacy in patients with advanced sarcoma is ongoing (NCT04995003).

Combinations of T-BsAb and PD-1/PD-L1 blockades have also been studied. T-BsAb upregulates PD-1 on T cells and PD-L1 on tumor cells, and a combination of anti-CEA (carcinoembryonic antigen) BsAb and PD-L1 inhibitor improved anti-tumor efficacy by increasing the frequency of TILs when compared with each monotherapy in preclinical models [146]. Anti-GD2 BsAb upregulated PD-1 on T cells and PD-L1 on osteosarcoma tumor cells. Sequential combination of PD-1 or PD-L1 inhibitor enhanced GD2-BsAb-driven T cell infiltration and survival of mice, and the tumor-suppressing effect was most effective when anti-PD-1 or anti-PD-L1 antibody treatment was prolonged [62]. But the ATC and ICI combinations have encountered key limitations including the short half-life of ICIs, requiring multiple administrations, inconsistent tumor penetration by T cells, and the risk of systemic on-target off-tumor toxicities [142,147]. Despite these concerns, clinical trials to test the efficacy of combining ATC with ICIs are recruiting patients with ALL (NCT05310591), sarcomas (NCT04995003), or relapsed/refractory Hodgkin lymphoma (NCT04134325).

Targeting TME is another promising strategy to overcome the limitations of T cell immunotherapy, addressing both the physical barriers and immune-hostile TME of osteosarcoma [95,99,148]. The majority of CAR T cell therapies have struggled with dose-limiting toxicities and poor efficacy against solid tumors. CAR T cells could not efficiently infiltrate the TME, requiring intratumoral injection to exert tumoricidal effects [11,65], and even then unable to survive in the immunosuppressive and hypoxic TME. In addition, CAR T cells or BsAb-driven T cells themselves recruited even more MDSCs and M2 macrophages into tumors [74], compromising their own efficacy [73]. While M2 macrophages are consistently correlated with fewer TILs, lung metastasis, and poor prognosis [149,150,151], the shift from M2 to M1 phenotype induced the regression of metastatic lesions [148,152]. All-trans retinoic acid (ATRA) is known to restrict osteosarcoma initiation and prevent metastasis by suppressing MDSCs and M2 polarization of macrophages, as well as disrupting TAM-cancer stem cell pathways [73,153,154]. Combined therapy of GD2-CAR T cells with ATRA significantly improved anti-tumor efficacy against sarcoma xenografts [73]. Trabectedin also reprograms the TME by targeting macrophages and monocytes, thereby inhibiting osteosarcoma tumor growth and lung metastases. Trabectedin combined with the PD-1 inhibitor significantly enhanced the number of CD8(+) TILs, improving treatment efficacy against osteosarcoma [155]. MDSCs have also been shown to affect the potency of CAR T cells, where targeting tumor MDSCs by anti-Gr1, anti-GM-CSF, or anti-PD-L1 antibody has improved treatment efficacy of anti-CEA CAR T cells in colon cancers [156]. In similar studies, TME modulation has greatly improved outcomes of BsAb-based T cell immunotherapy, where anti-Gr-1, anti-Ly6G, or anti-Ly6C antibodies to deplete MDSCs or clodronate liposome or anti-CSF1R antibodies to deplete TAMs were effectively combined with GD2-EATs or HER2-EATs for treating osteosarcoma [147]. MDSC depletion facilitated EAT trafficking and infiltration into osteosarcoma, resulting in improved tumor control. Depletion of TAMs was more effective than MDSC depletion to drive T cells into tumors, inducing more potent in vivo anti-osteosarcoma response. In these studies, dexamethasone before GD2-EAT injection predominantly depleted monocytes in the blood and macrophages in tumors while promoting GD2-EAT infiltration and anti-tumor activity [147].

Besides ICI and TME modulation, targeting neovasculature is yet another potential strategy to improve the efficacy of T cell immunotherapy in osteosarcoma. Bevacizumab combined with chemotherapy has been FDA approved as first-line therapy in multiple cancers including colorectal carcinoma, non-small cell lung cancer, ovarian carcinoma, breast cancer, and hepatocellular carcinoma [157,158,159]. Combination of anti-angiogenic therapy and ICI normalizes the vascular–immune cross-talk to potentiate cancer immunity, becoming a compelling combination strategy in clinical trials [160]. Bevacizumab has proven synergistic effects with the PD-1 inhibitor in advanced renal cell carcinoma and hepatocellular carcinoma [161,162], and many clinical trials are testing the efficacy of ICIs plus anti-angiogenic agents in a variety of solid cancers [50]. The addition of VEGF blockade to T-BsAb has shown promising synergy against osteosarcoma. Anti-VEGF (bevacizumab) or anti-VEGFR2 antibodies (DC101) significantly enhanced the trafficking of EATs into tumors and CD8(+) T cell infiltration, improving the in vivo anti-tumor effect. VEGF blockade normalized tumor vasculature by inducing high endothelial venules (HEVs) and increased CD8(+) TIL survival and dispersion while mitigating the immunosuppressive TME. These findings suggest that the VEGF pathway plays a key role in developing the immune-hostile TME of osteosarcoma, and targeting the VEGF/VEGFR2 pathway is an effective strategy to overcome TME and improve the clinical efficacy of T cell immunotherapy in osteosarcoma [163].

## 6. Conclusions

With the arrival of T cell immunotherapy, ICIs and ATC using BsAb or CAR may provide alternative options for relapsed or refractory osteosarcoma. In order to realize the true potential of T cell immunotherapy, osteosarcoma TME needs attention: a dense ECM, high densities of M2 macrophages and MDSCs, and abnormal tumor angiogenesis, which create hypoxic and acidic tumor environment that sabotage T cell immune responses. Targeting TME using the VEGF blockade, TAMs or the MDSC modulation or softening ECM may provide promising options to overcome these hurdles for T cell immunotherapy in osteosarcoma. By combining small molecule inhibitors with no cross-resistance or toxicities, curing osteosarcoma may be even possible if these strategies can be adopted upfront before tumors develop pan-resistance.

## Figures and Tables

**Table 1 ijms-24-12520-t001:** Clinical trials of immune checkpoint inhibitors (ICIs) in osteosarcoma.

Clinical Trial	Target	Eligibility (Age)	Phase	Status	Reference
Pembrolizumab in patients with advanced sarcomas	PD-1 inhibitor	>12	phase 2	completed	NCT02301039 [43,51]
Nivolumab with or without ipilimumab in treating recurrent or refractory solid tumors or sarcomas	PD-1 inhibitor + CTLA-4 inhibitor	>1	phase 1 and 2	active, not recruiting	NCT02304458 [52]
Nivolumab with or without ipilimumab for patients with metastatic sarcoma	PD-1 inhibitor + CTLA-4 inhibitor	>18	phase 2	active, not recruiting	NCT02500797 [46]
Atezolizumab in pediatric and young adult patients with solid tumors	PD-L1 inhibitor	<30	phase 1 and 2	terminated	NCT02541604 [53]
Durvalumab plus tremelimumab in multiple sarcoma subtypes	PD-L1 inhibitor + CTLA-4 inhibitor	>18	phase 2	active, not recruiting	NCT02815995 [47]
Avelumab in patients with recurrent or progressive osteosarcoma	PD-L1 inhibitor	12–49	phase 2	active, not recruiting	NCT03006848
Pembrolizumab in patients with relapsed or metastatic osteosarcoma	PD-1 inhibitor	>18	phase 2	terminated	NCT03013127 [44]
Nivolumab plus ABI-009 (nab-sirolimus) for advanced sarcoma and certain cancers	PD-1 inhibitor	>12	phase 1 and 2	completed	NCT03190174
Sunitinib and/or nivolumab plus chemotherapy in advanced soft tissue and bone sarcomas	PD-1 inhibitor	20–80	phase 1 and 2	recruiting	NCT03277924 [54]
Apatinib plus camrelizumab for advanced osteosarcoma	PD-1 inhibitor		phase 2	completed	NCT03359018 [50]
Nivolumab plus ipilimumab in non-resectable sarcoma and endometrial carcinoma	PD-1 inhibitor + CTLA-4 inhibitor	>18	phase 2	unknown	NCT03449108
Nivolumab and azacitidine for recurrent, resectable osteosarcoma	PD-1 inhibitor	<39	phase 1 and 2	recruiting	NCT03628209
Socazolimab in high-grade osteosarcoma	PD-L1 inhibitor	18–55	phase 1 and 2	recruiting	NCT03676985
Famitinib plus camrelizumab and famitinib alone and famitinib plus ifosfamide in advanced osteosarcoma	PD-1 inhibitor	>12	phase 1 and 2	withdrawn	NCT04044378
Multi-antigen autoimmune cell injection (MASCT-I)combined with apatinib and/or anti-PD1 antibody in the treatment of tissue sarcoma	PD-1 inhibitor	14–70	phase 1	active, not recruiting	NCT04074564
Camrelizumab in combination with neoadjuvant chemotherapy in osteosarcoma	PD-1 inhibitor	14–65	phase 2	recruiting	NCT04294511
MAPI + camrelizumab vs. API + apatinib vs. MAPI in patients with a poor response to preoperative chemotherapy for newly diagnosed high-grade osteosarcoma	PD-1 inhibitor	>12	phase 2	unknown	NCT04351308
Socazolimab for maintenance therapy in patients with high-grade osteosarcoma	PD-L1 inhibitor	>12	phase 3	not yet recruiting	NCT04359550
Niraparib and dostarlimab in pediatric participants with solid tumors	PD-1 inhibitor	0.5–17	phase 1	recruiting	NCT04544995
Oleclumab and durvalumab for the treatment of recurrent, refractory, or metastatic sarcoma	PD-L1 inhibitor	>12	phase 2	recruiting	NCT04668300
Bempegaldesleukin (BEMPEG: NKTR-214) in combination with nivolumab in recurrent or treatment-resistant cancer	PD-1 inhibitor	<30	phase 1 and 2	terminated	NCT04730349
Regorafenib and nivolumab in refractory or relapsed osteosarcoma	PD-1 inhibitor	>5	phase 2	active, not recruiting	NCT04803877
Atezolizumab and cabozantinib for recurrent or metastatic Osteosarcoma	PD-L1 inhibitor	>12	phase 2	not yet recruiting	NCT05019703
Combination of pembrolizumab and cabozantinib in patients with advanced sarcomas	PD-1 inhibitor	>18	phase 2	recruiting	NCT05182164
Tislelizumab combined with chemotherapy in bone metastatic sarcoma	PD-1 inhibitor	>18	phase 2	recruiting	NCT05241132
Neoadjuvant dual checkpoint inhibition and cryoablation in relapsed/refractory pediatric solid tumors	PD-1 inhibitor + CTLA-4 inhibitor	>1	phase I and 2	recruiting	NCT05302921

**Table 2 ijms-24-12520-t002:** Clinical trials of adoptive T cell therapy for osteosarcoma.

Clinical Trial	Target	Eligibility (Age)	Phase	Status	Reference
HER2 chimeric antigen receptor expressing T cells in advanced sarcoma	HER2-CAR T cells	Child, adult, older adults	phase 1	active, not recruiting	NCT00902044 [64]
Humanized 3F8 bispecific antibody (Hu3F8-BsAb) in relapsed/refractory neuroblastoma, osteosarcoma and other solid tumor cancers	Humanized 3F8 bispecific antibody	1–17	phase 1 and 2	terminated	NCT03860207
iC9-GD2-CAR-VZV-CTLs in refractory or metastatic GD2-positive sarcoma and neuroblastoma	GD2 T cells, VZV vaccine, CPM, fludarabine	child, adult, older adult	phase 1	active, not recruiting	NCT01953900 [92]
Haploidentical transplant and donor NK cells for solid tumors	Donor NK cell	child, adult, older adult	phase 2	active, not recruiting	NCT02100891
T cells expressing an anti-GD2 CAR in GD2+ solid tumors	GD2-CAR T cell	1–35	phase 1	completed	NCT02107963
Activated T cells armed with GD2 bispecific antibody in neuroblastoma and osteosarcoma	GD2 BsAb armed T cell	1–29	phase 1 and 2	unknown	NCT02173093 [91]
Aβ CD19+ depleted haploidentical transplantation plus zometa for pediatric hematologic malignancies and solid tumors	TCRαβ+/CD19+ depleted haploidentical stem cells	0.5–21	phase 1	recruiting	NCT02508038
Treatment of relapsed or refractory neuroblastoma and osteosarcoma with expanded haploidentical NK Cells and Hu14.18-IL2	Ex vivo expanded and activated haploidentical donor NK cells, and Hu14.18-IL2	0.5–26	phase 1	withdrawn	NCT03209869
Fourth-generation safety-engineered CAR T cells targeting sarcomas	Sarcoma-specific CAR-T cells	1–75	phase 1 and 2	recruiting	NCT03356782
Anti-GD2 CAR T cells in high-risk and/or relapsed/refractory neuroblastoma or other GD2-positive solid tumors	GD2-CART 01	1–25	phase 1 and 2	recruiting	NCT03373097
LN-145 or LN-145-S1 in relapsed or refractory cancers	Autologous tumor infiltrating lymphocytes LN-145 or LN-145-S1	16–70	phase 2	recruiting	NCT03449108
NY-ESO-1-specific T cell receptor (TCR) T cell in sarcoma	NY-ESO-1(TCR affinity enhancing T cell	14–70	phase 1	recruiting	NCT03462316
EGFR806 CAR T cell immunotherapy for recurrent/refractory solid tumors	EGFR 806CAR(2G)-EGFRt, CD19CAR(2G)-T2A-HER2tG	1–30	phase 1	recruiting	NCT03618381
C7R-GD2.CAR T cells for patients with relapsed or refractory neuroblastoma and other GD2-positive cancers	C7R-GD2.CART cells	1–74	phase 1	recruiting	NCT03635632
Study of CAR T cells targeting the GD2 with IL-15+i caspase 9 for relapsed/refractory neuroblastoma or relapsed/refractory osteosarcoma	iC9.GD2.CAR.IL-15 T cells	>1.6	phase 1	recruiting	NCT03721068
Combination immunotherapy targeting sarcomas	Multiple sarcoma-specific CAR-T cells and sarcoma vaccines	1–75	phase 1 and 2	recruiting	NCT04433221
B7H3 CAR T cell immunotherapy for recurrent/refractory solid tumors	B7H3-specific CAR, bispecific B7H3xCD19 CAR	<26	phase 1	recruiting	NCT04483778
GD2-targeted modified T cells (GD2CART) in relapsed/refractory osteosarcoma and neuroblastoma	GD2-CAR T cell	<35	phase 1	suspended	NCT04539366
Clinical study of CD276 targeted CAR T cell in CD276-positive advanced solid tumor	CD276 CAR T cells	1–70	phase 1	not yet recruiting	NCT04864821
B7-H3-CAR T cell therapy for pediatric patients with solid tumors	B7-H3 CAR T cell	<21	phase 1	recruiting	NCT04897321
HER2 CAR T cells in combination with checkpoint blockade in patients with advanced sarcoma	HER2-CAR T cell + pembrolizumab or nivolumab	1–25	Phase 1	recruiting	NCT04995003
Fluorescein-specific (FITC-E2) CAR T cells in combination with folate–fluorescein (UB-TT170) for osteogenic sarcoma	SCRI-E2CAR_EGFRtv1 + UB_TT170	15–30	phase 1	recruiting	NCT05312411
T cell membrane-anchored tumor targeted IL12 (Attil12)-T cell therapy in advanced/metastatic soft tissue and bone sarcoma	autologous tumor-targeted IL12 T cells	>12	phase 1	not yet recruiting	NCT05621668
CAR.70-engineered IL15-transduced cord blood-derived NK cells in conjunction with lymphodepleting chemotherapy for advanced renal cell carcinoma, mesothelioma and osteosarcoma	CAR.70/IL15-transduced CB-derived NK cells	18–80	phase 1 and 2	not yet recruiting	NCT05703854

## Data Availability

Data are contained within the article.

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
