# Peer review of "Promise and Challenges of T Cell Immunotherapy for Osteosarcoma"

_ijms, 2023, doi:10.3390/ijms241512520_

Round 1

Reviewer 1 Report

I have gone through the manuscript. Topic is interesting but it needs moderate conditioning. PD-L1/PD-1 signaling has not been properly explained. 

Authors should systematically characterize immunological advancements in animal model studies and clinical trials. These aspects have to be addressed very comprehensively. Clinical trials should be discussed in a separate section with special focus on the existing knowledge gaps. How the strategies can be re-visited to maximize the gains. Likewise, animal model studies should be summarized more carefully. How metastasis can be pharmaceutically targeted should be explained in a detailed manner in context of PD-L/PD signaling

Moderate language editing is required. 

Author Response

I have gone through the manuscript. Topic is interesting but it needs moderate conditioning. PD-L1/PD-1 signaling has not been properly explained.

→The PD-1/PD-L1 signaling pathway in osteosarcoma is explained in the

Authors should systematically characterize immunological advancements in animal model studies and clinical trials. These aspects have to be addressed very comprehensively. Clinical trials should be discussed in a separate section with special focus on the existing knowledge gaps. How the strategies can be re-visited to maximize the gains. Likewise, animal model studies should be summarized more carefully. How metastasis can be pharmaceutically targeted should be explained in a detailed manner in context of PD-L/PD signaling.

→ We have revised the manuscript to address animal data and clinical studies comprehensively. Clinical data of ICIs for osteosarcoma is summarized in a separate table (Table 1). We explicitly distinguished preclinical data from clinical trial results by separating them into different paragraphs. In addition, we clarified the concept detailing how the metastasis of osteosarcoma could be targeted by PD-1/PD-L1 blockades, as evidenced by PD-1 inhibitors which effectively control pulmonary metastasis by increasing CD4(+) and CD8(+) TILs and enhancing the cytolytic activity of CD8(+) T cells in the lung [41]. Both human and murine metastatic osteosarcomas express the PD-L1, which could functionally impair tumor-infiltrating CTLs by engaging their surface PD-1. This model was supported by studies where PD-L1 blockade improved the function of osteosarcoma TILs in vivo, decreasing tumor burden and increasing survival of mice carrying metastatic osteosarcoma [39].

Reviewer 2 Report

This review provides a comprehensive overview of the state of the art in the use of T-cell immunotherapy against osteosarcoma, the most common cancer affecting bone tissue. Nowadays, this is a hot topic as immunogenic T-cell therapy has not been successfully used in the treatment of sarcomas.

The review presents a comprehensive overview of the most recent literature on the topic, focusing on the challenges that the researchers will face to make the use of T-cell immunotherapy effective in the treatment of human osteosarcoma.

Minor revision:

Page 1 – row 29: Please substitute percent with %

Page 2 – row 71-73. This phrase is not clear, please rephrase

Page 3 – 141-147. The phrase is too long and not easy to follow.

Page 5 – 174-178. Not clear

Figure1: I suggest to add detailed legend of the figure.

Author Response

This review provides a comprehensive overview of the state of the art in the use of T-cell immunotherapy against osteosarcoma, the most common cancer affecting bone tissue. Nowadays, this is a hot topic as immunogenic T-cell therapy has not been successfully used in the treatment of sarcomas.

The review presents a comprehensive overview of the most recent literature on the topic, focusing on the challenges that the researchers will face to make the use of T-cell immunotherapy effective in the treatment of human osteosarcoma.

Minor revision:

Page 1 – row 29: Please substitute percent with %

→ Corrected

Page 2 – row 71-73. This phrase is not clear, please rephrase

→ The phrase was rephrased. “This osteosarcoma-specific TME impedes T cell infiltration into tumors, accelerates immune effector cells’ exhaustion and anergy, and derails anti-tumor immunity, creating both a major roadblock as well as a potential tumor vulnerability.”

Page 3 – 141-147. The phrase is too long and not easy to follow.

→ Edited.

Several cases reported that the combination of anti-CTLA4 and anti-PD-1 antibodies induced remission and tumor stabilization in patients with metastatic osteosarcoma [48, 49], while the addition of camrelizumab (anti-PD-1 inhibitor) to the inhibition of vascular endothelial growth factor receptor 2 (VEGFR2)] using apatinib [tyrosine kinase inhibitor (TKI)] was shown to prolong PFS of patients with advanced osteosarcoma compared with apatinib alone [50].

Page 5 – 174-178. Not clear

→ Edited.

Despite its HER2 antigen density being too low for conventional IgG-mediated cytotoxicity, osteosarcoma was effectively killed by HER2-CAR T cells, which when injected intratumorally induced regression of established osteosarcoma xenografts, prolonging survival of the mice.

Figure1: I suggest to add detailed legend of the figure.

  • The legend was below the figure. We now moved the legend above the figure.

Reviewer 3 Report

 The review by Park et al covers the current status of T cell immunotherapies for osteosarcoma, summary of the challenges, and combination strategies. The manuscript is well written, thorough, and appropriate for publication in the international journal of molecular sciences.  

Minor points:

1.     Figure 1 was not mentioned in the text.

2.     Figure 1 needs a description for schematic diagram. 

3.     Line 281, there is typo “express”

Author Response

The review by Park et al covers the current status of T cell immunotherapies for osteosarcoma, summary of the challenges, and combination strategies. The manuscript is well written, thorough, and appropriate for publication in the international journal of molecular sciences.  

Minor points:

  1. Figure 1 was not mentioned in the text.

→ Figure 1 was mentioned on page 9, line 260.

  1. Figure 1 needs a description of the schematic diagram. 

→ The legend was below the figure. We have moved the legend above the figure. 

  1. Line 281, there is typo “express”

→ Corrected.

Round 2

Reviewer 1 Report

It looks in good form now. 

Minor spell check and editing required.